# Strategies for Improving Vascularization in Kidney Organoids: A Review of Current Trends

**DOI:** 10.3390/biology12040503

**Published:** 2023-03-26

**Authors:** Ran Konoe, Ryuji Morizane

**Affiliations:** Massachusetts General Hospital, Harvard Medical School, Boston, MA 02114, USA

**Keywords:** kidney organoid, vascularization, iPSCs, ECs, maturation, organ on chip, nephron

## Abstract

**Simple Summary:**

Organoids are small, 3D models of organs that can help researchers study organ function and disease. Kidney organoids have the potential to treat chronic kidney disease, which affects over 800 million people. However, creating fully functioning kidney organoids is still a challenge due to functional immaturity and the lack of blood vessels and tissue organization. This article discusses the current challenges and efforts in developing matured and vascularized organoids.

**Abstract:**

Kidney organoids possess the potential to revolutionize the treatment of renal diseases. However, their growth and maturation are impeded by insufficient growth of blood vessels. Through a PubMed search, we have identified 34 studies that attempted to address this challenge. Researchers are exploring various approaches including animal transplantation, organ-on-chips, and extracellular matrices (ECMs). The most prevalent method to promote the maturation and vascularization of organoids involves transplanting them into animals for in vivo culture, creating an optimal environment for organoid growth and the development of a chimeric vessel network between the host and organoids. Organ-on-chip technology permits the in vitro culture of organoids, enabling researchers to manipulate the microenvironment and investigate the key factors that influence organoid development. Lastly, ECMs have been discovered to aid the formation of blood vessels during organoid differentiation. ECMs from animal tissue have been particularly successful, although the underlying mechanisms require further research. Future research building upon these recent studies may enable the generation of functional kidney tissues for replacement therapies.

## 1. Introduction

Organoids are three-dimensional, multicellular cultures in vitro that closely mimic the tissue structure and function of their corresponding organs [1,2]. Over the past decade, the organoid has become a valuable tool in the study of organogenesis, genetic diseases, and infections. In earlier work, kidney organoids were generated from pluripotent stem cells (PSCs) including embryonic stem cells (ESCs) and induced pluripotent stem cells (iPSCs) [2]. By following in vivo organogenesis, directed differentiation approaches were developed to generate kidney organoids from PSCs [2,3,4]. Kidney organoids contain varied cell types, forming the nephron structure, a kidney functional unit, and they are effectively used in the studies of kidney development, genetic diseases, and infections [5,6,7]. More recent studies have shown the creation of tubular organoids (tubuloids) using the primary culture of adult human kidneys, expanding the applications of kidney organoids to varied disease studies [8,9,10]. Tubuloids can be developed by relatively simple procedures and are expected to be useful for personalized medicine and examining the therapeutic efficacy for each patient [11].

However, both PSCs and adult cell-derived organoids have limitations in their size due to the lack of perfusable blood vessels [12,13,14,15]. In human bodies, organs receive essential nutrients and oxygen through vascular networks, whereas kidney organoids lack such central circulation systems and rely on molecular diffusion for nutritional delivery. The diffusion range is limited without perfusable vessels, restricting the maximum size of kidney organoids in vitro. Furthermore, the absence of the vascular system impairs the function of the kidneys whose primary functions are blood filtration and urinary excretion for homeostasis. Kidneys require complex and dense vascular networks for glomerular and tubular functions. To achieve functional kidney organoids for restoring kidney function, it is crucial to have a well-developed blood supply. Without adequate blood circulation, the kidney filtration process in the organoids is severely impaired.

In this review, we summarize recent studies that attempted to improve the vascularization and maturation of kidney organoids. We will also discuss the potential directions toward the ultimate goal of generating functional kidneys with a focus on factors involved in vascularization.

## 2. Materials and Methods

Articles were searched and collected from PubMed using the following keywords: (organ-on-chip OR microfluidics OR vascularization OR maturation) AND (kidney organoid OR nephron organoid OR renal organoid). Using the described keywords, 225 studies were found in PubMed. From these 225 studies, 191 publications of review, commentary, and interview reports were excluded. We then selected the studies which are related to the vascularization and maturation of human kidney organoids. The remaining 34 studies were categorized into 4 groups based on their methods of organoid vascularization: (1) animal models, (2) organ on chip (OoC), (3) extracellular matrix (ECM), and (4) others. We added several independent articles to reference in the discussion and introduction. 

## 3. Result

### 3.1. Animal Model

Organoid transplantation into animals is a widely utilized method to enhance the vascularization and maturation of kidney organoids. This approach offers benefits in terms of organoid vascularization, as the animal provides more physiological environments than in vitro, including a well-functioning circulation system. If a human-origin kidney organoid is transplanted into an animal with intact immune function, it is recognized as an alien entity by the host’s immune system, triggering a robust immune response that leads to the elimination of the organoid. To prevent this immune rejection, immunodeficient animals, such as BALB/c nude mice and NOD/SCID mice, are utilized. The most frequently used site of transplantation is under the renal capsule in non-obese diabetic/severe combined immunodeficiency (NOD/SCID) mice or athymic nude rats [16,17,18]. Van den Berg et al. transplanted hiPSC-derived human kidney organoids into kidney capsules of NOD/SCID mice [12]. Before transplantation, hiPSCs were subjected to 3D culture for a period of 18 days to form kidney organoids, after seven days of cultivation in monolayer. The dimensions of the organoid after 28 days of culture in kidney capsules of NOD/SCID mice exhibited a progressive augmentation in tandem with the duration of transplantation. Additionally, following the transplantation, the application of toluidine blue staining demonstrated the existence of glomerular and tubular structures within the organoid. Analysis using Scanning Electron Microscopy (SEM) indicated the existence of vasculature within the organoid, as well as glomerular structures. In fact, transplanted hiPSC-derived organoids contained host-derived mouse endothelial cells (MECA-32+). Immunostaining enabled us to determine that host cells infiltrated glomerular-like structures (NPHS1+ and WT1+), while peritubular vascularization was also detected in connection with tubular epithelium. Comprehensive TEM images of the glomerular and tubular structures demonstrated maturity-associated characteristics such as the formation of foot processes, glomerular basement membranes, and the presence of microvilli. Nevertheless, these structures appeared relatively disorganized when compared to native kidneys. 

Zhang et al. cultured organoids in vivo for 14 days, revealing the presence of glomerular and tubular structures. [16]. However, they also noted widely observed areas of poor differentiation and osteo differentiation, with very sparse and poorly differentiated regions in organoids in vitro. In comparison to prior research conducted by van den Berg et al., discrepancies were observed in the methods of animal surgery, in vitro induction, and in vivo culture duration. Notably, Zhang et al. confirmed that the grafts (kidney organoids) were sustained by the internal environment. Nonetheless, the shorter in vivo culture periods utilized by Zhang et al. failed to promote adequate differentiation and maturation of kidney compartment cells in contrast to those of van den Berg et al. Moreover, kidney organoids generated by a protocol modified from Takasato’s differed significantly, with the tubules of kidney organoids being considerably sparser than those of normal kidneys. Such observations indicate that the quality of organoid culture prior to transplantation can impact post-transplantation growth. Therefore, future studies ought to optimize induction conditions to enhance the efficiency of kidney organoid induction. Validation is also required for the optimal animal species, transplantation sites, and organoid sizes before transplantation, to prolong the in vivo culture period. Moreover, pre-transplantation unilateral nephrectomy is expected to augment blood flow to the remaining kidney, thereby likely fostering kidney organoid maturation. Compensatory hypertrophy of the uninvolved kidney is a frequent clinical phenomenon following unilateral nephrectomy [19,20]. To scrutinize the effect of unilateral nephrectomy on the engraftment of kidney organoids into the sub-kidney capsule, Zhang et al. transplanted human pluripotent stem cell-derived kidney organoids. The organoids were initially cultured under static conditions for 7 days, followed by 19 days of 3-dimensional culture, though the timing of the transplantation was not specified in this study. The experimental group was comprised of mice subjected to unilateral nephrectomy, while the control group was not. After a two-week incubation period, statistical analyses demonstrated a significantly greater volume of kidney organoids in the experimental group compared to the control group [16].

Lymph nodes have been used as an alternative transplantation site for mouse metanephric kidneys. Francipane et al. transplanted human fetal kidneys and hiPSC-derived kidney organoids into mice jejunal lymph nodes. After one week of fetal kidney transplantation, host endothelial infiltration was observed, and human glomeruli were entirely revascularized by mouse endothelia by 8 weeks of transplantation [21]. The hiPSC-derived kidney organoids were generated by the Takasato protocol [2]. In one week after transplanting organoids, they were successfully engrafted and showed signs of host endothelial infiltration. Mouse CD31+ cells were found in WT-1+ aggregates, and LTL+ tubules were observed. Glomerular-like structures containing mouse endothelia and podocytes were detected adjacent to presumptive proximal tubules, six weeks after transplantation. Notably, cartilage formation was often observed as a byproduct of hiPSC differentiation. Hepatic, pancreatic, and thymic cells have also shown similar maturation in the lymph nodes [22]. It remains to be verified whether this maturation occurs in other sites of lymph nodes that are close to the body surface.

While kidney capsules can be used for kidney organoid transplantation, the surgical procedure requires a laparotomy, a major surgery. Hence, some studies tested other sites for organoid transplantation (Table 1). Kaisto et al. have successfully vascularized kidney organoids by implanting organoids into the chorioallantoic membrane (CAM) of a chicken [18,23]. The CAM is a transparent and thin tissue that enables shorter operative time and real-time imaging. However, after five days of incubation, the CAM begins to deteriorate, rendering this method unsuitable for extended incubation. In contrast, chicken embryos can continue to develop normally for up to 15 days in immunocompromised conditions after their eggshell has been opened. [24]. Koning et al. successfully transplanted human kidney organoids into the coelomic cavity of chicken embryos, where the organoids expanded vascular epithelial cells to create a chimeric vascular network that merged with host-derived blood vessels [24]. 

Based on these studies, the cell number of transplanted organoids appears to be important for successful experiments. Cell counts below 10,000 were reported to result in graft atrophy after implantation [17]. The differentiation stage of organoids is also another key factor, and more differentiated and matured organoids appear to improve their survival after transplantation. The secretion factors such as the vascular endothelial growth factor (VEGF) from the grafts likely play an essential role in building vascular networks between donor and host cells, supporting graft survival [12]. Post-transplanted organoids demonstrated comparable filtration functions with some levels of selectability based on varied sizes of dextran [12,26]. However, the degrees of glomerular vascularization and organoid survival are yet to be improved in future studies. 

### 3.2. Organ on Chips

An organs-on-chip (OoC) is a system that contains artificial or primary miniature tissues cultured within microfluidic chips [27]. Compared to conventional cell culture models, OoC better recapitulates the cellular microenvironment that may be requisite for the maturation and vascularization of organoids. In vitro models using the OoC are better suited for real-time observation and intervention with larger numbers of factors than in vivo animal models. Furthermore, the human kidney organoid model can be scaled up for high-throughput screening at low costs without animals [28]. However, several challenges remain for the culturing organoids in the OoC. Although marked improvement has been shown compared to organoids cultured with other techniques, organoid size, vascularization, and maturation still have not reached ideal levels [29]. In this section, we will introduce the recent progress in the OoC and the key factors for organoid growth.

The biochemical factors involved in organoid maturation have long been investigated [30,31]. However, it is unclear whether the molecular concentrations of these factors are optimal during the differentiation phase of kidney cells and whether these molecules induce cell fate decisions directly, synergistically, or through a paracrine mechanism. It is also unclear whether a particular combination of factors can unbalance differentiation into specific subtypes of the developing kidney and how this is regulated over time. The combinations are so enormous that a method to investigate them over time in static culture is a daunting task. In a massively organized and systematic approach to studying the impact of factorial combinations of these small molecules, growth factors, and paracrine signaling events on kidney cell fate, Glass et al. developed a micro-bioreactor arrays (MBA) platform that can apply fluid flow [30]. Their novel approach, using over 1000 unique biochemical input conditions from four different MBA-based experiments, allowed the rapid assessment of the relative impact of these numerous varieties of input conditions on the expression of markers of kidney cell lineages. The actual steps of human development occur through the intricate interplay of neighboring organs. Similarly, the growth and differentiation of organoids are accompanied by arduous stages of development, requiring the precise and careful addition, and sometimes cessation, of various factors implicated in their differentiation. To explore the relationship between growth factors and their effects on organoid development, including those that have yet to be discovered, this device that enables the simultaneous multifactorial manipulation of the addition and cessation of external factors may prove to be a promising tool.

The study of these combinations of biochemicals has yielded fruitful results, but biochemical stimuli alone were insufficient for the complete maturation of the kidney organoids. Recent studies have revealed that biomechanical factors from the micro niche of cells change gene expression via epigenetic decoration [32,33,34]. Homan et al. constructed a simple millifluidic culture system to investigate the effects of extracellular matrix (ECM), medium composition, fluidic shear stress (FSS), and co-culture with human endothelial cells on the in vitro development of vascularized kidney organoids (Figure 1) [14]. The results showed that kidney organoids cultured under flow had more mature podocytes, tubular compartments, enhanced cell polarity, and mature gene expression compared to static cultures. These results suggest that FSS is an important mechanical factor in the tissue microenvironment that promotes vascularization and maturation of kidney organoids. To optimize the shape of the OoC, Lee et al. simulated the shear stresses and the fluid velocity distribution acting on the surface of the organoids, which are involved in vessel formation [31]. This study showed that changing the internal configuration of the OoC can control the shear stress applied to a specific site of organoids. They have also revealed the combination of vascular endothelial growth factor (VEGF) with other biomechanical stimuli can synergistically improve the vascularization of kidney organoids. Furthermore, Menéndez et al. reported that co-culture of human umbilical vein endothelial cells (HUVECs) with kidney organoids in a perfusable microfluidic organ on chip leads to endothelial cell migration and vascularization in the kidney organoids [35].

As these studies show, the advantages of the OoC are the insertion of appropriate biomechanical stress, biochemical factors, and cells that provoke the vascularization of organoids. We expect further research to optimize these external factors to further improve the maturity of organoids.

### 3.3. ECM & Others

Here, we will introduce various other studies which utilized ECMs and other growth conditions different from the above sections. These studies focused on variables that have not been previously studied in depth and yielded impactful results that encourage further exploration. 

In the organoid field, various hydrogels have been investigated to influence cell behavior. Ruiter et al. cultured kidney organoids in three hydrogels with adjustable stiffness (ranging from 0.1 to 20 kPa) and two soft hydrogels with different stress relaxation to investigate the effect of hydrogels on organoid maturation [36]. Organoid maturation was investigated by the expression of off-target ECM and epithelial-mesenchymal transition (EMT)-related markers and variation in the formation of the lumen and ciliary structures in kidney organoids cultured with each hydrogel. The results showed that kidney organoids encapsulated in soft hydrogels with rapid relaxation properties matured more by the above parameters compared to hard or loose hydrogels. Garreta et al. showed that soft hydrogels promoted the expression of genes involved in embryonic and mesoderm differentiation in the undifferentiated state compared to hard hydrogels [23]. These results underline the possibility of modulating hydrogel properties to influence organoid maturation and function.

Hydrogel extracellular matrix (ECM) plays a crucial role in providing mechanical support and a biochemical microenvironment that supports cell proliferation and differentiation. In a study by Kim et al. [37], PSC-derived kidney organoids cultured in hydrogels that were prepared using decellularized kidney ECM (dECM) exhibited a substantial vascular network with their own endothelial cells. Single cell transcriptomics analysis revealed that kidney organoids cultured in dECM exhibited a more advanced pattern of glomerular development, exhibiting similarities to human kidney structures compared to those cultured without dECM. This effect was associated with the presence of VEGF in the dECM. Interestingly, even when VEGF in the dECM hydrogels was suppressed using bevacizumab, a VEGF inhibitor, more vascular development was observed in dECM cultures. These intriguing observations suggest that dECM has factors beyond VEGF that enhance organoid angiogenesis.

Many previous experiments have used oxygen concentrations similar to atmospheric levels (about 21%) which is standard in most cell culture environments. However, since kidneys in vivo develop under hypoxic conditions, Schumacher et al. sought to replicate this in vivo hypoxia by cultivating kidney organoids at 7% O_2_ for up to 25 days [38]. Following incubation, the organoids were assessed for kidney formation, growth factor expression, such as VEGF-A, and angiogenesis. Whole-mount imaging confirmed that culturing kidney organoids under hypoxic conditions resulted in a more homogeneous endothelial network morphology with enhanced sprouting and interconnections. A detailed analysis revealed increased expression of VEGFA-189 and VEGFA-121, and decreased expression of VEGF-A165b among the different splice variants of VEGF. VEGFA-189 and 121 are known to promote angiogenesis by facilitating cell migration. The expression of these growth factors may be promoted in a hypoxic environment, leading to increased vascularization of the organoids. However, the downregulation of VEGFA-165b in culture remains a mystery that requires further investigation. This study indicates that hypoxic culture conditions may enhance organoid angiogenesis.

Lastly, we present a unique study utilizing silk as a scaffold. Sutures made of silk have a well-established clinical history, and their safety for use in humans is widely recognized. Moreover, the in vivo degradation rate of silk can be adjusted to span several months to years. Capitalizing on these attributes, Gupta and colleagues seeded silk with kidney organoids derived from human-induced pluripotent stem cells (hiPSCs) [39]. On this material, they found that cells were able to differentiate into epithelial characteristic of the developing kidney and that these structures were maintained after transplantation under the sub-renal capsule of immunodeficient mice. Furthermore, the addition of VEGF to silk promoted vascular engraftment. It has been discovered that silk can be used to facilitate the growth of kidney tissue. However, the proliferation of stromal cells within the graft and the organization of tubular tissue remain problematic.

## 4. Discussion

With the optimization of vascularization, the growth of organoids can be significantly promoted. However, several critical challenges remain for the prospective clinical application of organoids.

Firstly, the drainage system of the collecting duct is not developed in the transplanted kidney organoids, as stated in previous studies [12,17,40]. Although the transplanted organoids can integrate with the host vascular network to receive nutrients and blood supply to the glomeruli and tubules, the distal end of the organoid’s nephron fails to connect with the host’s drainage system. Thus, even if hemofiltration occurs in the transplanted organoid, the glomerular filtrate cannot be excreted from the body. Furthermore, the immaturity of the nephron epithelium in the transplanted organoids is another concern. This is demonstrated by the persistent expression of PAX2 in the proximal tubules, indicating that the tubules remain immature [41]. Additionally, stromal cell overgrowth is another challenge, which can be attributed to the unstable blood supply to the graft. Insufficient specification of the posterior intermediate mesoderm during organoid differentiation may also be responsible for nephron degradation and stromal expansion. Further research is required to explore the factors and conditions involved in the formation of nephrons and their connection to collecting ducts. Once this mechanism is clarified, it may be possible to establish an integrated drainage system with the host kidney [8,10].

The second concern is chimerism in animal transplantation models. Although animals offer an in vivo environment for the angiogenesis of kidney organoids, chimeric grafts are also of great interest. Upon transplantation, organoids become vascularized with host-derived cells, resulting in chimeric tissue that includes animal cells [12,25,42]. However, this poses an immune response risk when transplanted into humans. HLA mismatch between donor and recipient hiPSC can be addressed by gene editing or hiPSC reprogramming from the host, but uncontrolled immune rejection can still occur due to species differences. In addition, chimerism can complicate disease research since humans and animals differ. Several strategies are being investigated to mitigate chimerism. Asynchronous differentiation of progenitor cells and the maturation of mixed kidney organoids in vivo showed that all tissue, except for endothelial cells, was donor-supplied. Pleniceanu et al. demonstrated that vasculogenic cells, which include human mesenchymal stromal cells (MSCs) and endothelial colony-forming cells (ECFC), led to the formation of a donor-derived vascular network when transplanted as a mixed sample [42]. Autologous transplantation, where the patient’s own cell-derived organoids are transplanted following successful liver regeneration, may also be effective in preventing chimerism in kidney transplantation [43]. 

The optimal approach to prevent chimerism is by completely maturing organoids in vitro without the utilization of animal models. In the OoC domain, advancements in photolithography and soft lithography techniques have facilitated the formation of intricate microfluidic channels [44,45]. Nevertheless, the effect of these complex, thread-like flow paths on organoid vascularization remains unknown. Future investigations may uncover unknown factors that are essential for organoid maturation within the OoC field. 

While the significance of the Extracellular Matrix (ECM) in vascularization is widely acknowledged in the tissue regeneration realm [46,47,48,49], there is a dearth of research focused on its application in organoids. Organoids exhibit unique properties, distinct from other regenerative tissues, in that they undergo substantial alterations in size and shape as they mature. In contrast, engineered soft tissues and encapsulated pancreatic islets, which are intended for transplantation, are pre-determined in shape and size and do not experience notable growth following vascularization. As cells become larger, they move further away from the blood vessels, necessitating distribution within a 100 to 200 μm radius from blood vessels for cell survival. Therefore, larger organoids require complex vascularization. Initially, the ECM surrounding organoids acts as a scaffold to support them, but after growth, it can impede angiogenesis and become a barrier. During kidney development in vivo, the ECM undergoes constant remodeling, production, and degradation [50]. Recreating such dynamic ECM changes in vitro is challenging and may be crucial for organoid vascularization. Hence, it may be advantageous to employ biodegradable materials or ECM of animal origin to promote organoid vascular formation. 

Thirdly, a crucial challenge is the insufficient number of nephrons. Despite the advancements in dialysis therapy, the glomerular filtration and clearance rate of dialysis therapy remains only 10% of normal kidney function. In view of the potential for future adjunctive therapy, it has been estimated that a minimum of 50,000 glomeruli must be cultured [51]. However, current organoid culture techniques fall short of generating a sufficient number of nephrons, necessitating the need to augment nephron density in pre-transplant samples. To accomplish this, 3D printers and polymer-based nephron sheets can be employed to increase nephron density and simultaneously mature the required number of nephrons. Lawlor et al. demonstrated the use of high-resolution 3D bioprinters to produce kidney organoids that contained a greater number of nephrons than manual protocols [52]. Moreover, Wiersma et al. developed a protocol to create nephron sheets containing up to 30,000 to 40,000 glomerular structures [51]. 3D bioprinting could be useful to improve organoid differentiation and culture with optimization of the tissue shape and size. However, the selection of biocompatible materials for use in 3D bioprinting is limited, thereby restricting the availability of optimal materials for culture [53,54,55,56]. Moreover, during the process of printing cells into 3D structures, some cells may incur damage or death, which ultimately results in decreased viability and function. This is especially true in the printing of larger tissues, which poses a greater risk of cellular damage due to the longer elapsed time. It is projected that cell viability will be enhanced with the repeated minimal printing of individual organoids or smaller structures, as opposed to the methodology of printing the entire kidney tissue as a whole. Further development of techniques to produce sufficient quantities of nephrons for kidney function is necessary.

Fourthly, the safety of long-term culture is a crucial consideration in the development of kidney organoids. Although the culture period and survival of kidney organoids can be prolonged by accelerated vascularization, the use of human pluripotent stem cells (hiPSCs) as the source of kidney organoids presents a potential safety risk due to their inherent tumorigenicity [57]. Notably, the differentiation of hiPSCs often leads to heterogeneous cell populations, and even a small number of undifferentiated hiPSCs (less than 10,000) can result in teratoma formation in vivo [58,59]. Furthermore, hiPSCs that have differentiated into other lineages may give rise to tumors or produce undesirable tissues after transplantation [60,61]. To evaluate the safety of transplanted kidney organoids, Nam and colleagues implanted hiPSC-derived kidney organoids into NOD/SCID mice and monitored them for up to 6 weeks [25]. Four weeks after transplantation, the presence of cartilage and cysts was observed within the kidney organoids, which gradually enlarged over time. In grafts with confirmed tumors, the development of normal nephrons was observed after transplantation, but the normal nephrons atrophied as the tumors grew. Importantly, these tumors were found to be of human origin, suggesting that they contained non-kidney or partially differentiated cells other than organoids in the pre-transplant stage. Thus, the presence of undifferentiated and differentiated hiPSCs in the pre-transplantation stage is a potential safety concern for human kidney organoid transplantation. In clinical settings, the transplantation of organoids of the same size as the actual kidney would require the transplantation of billions or more hiPSC-derived cells into the patient. Even the presence of 0.001% undifferentiated hiPSCs may be unacceptable in such cases; therefore, protocols aimed at minimizing undifferentiated hiPSCs are of utmost importance.

Currently, dialysis therapy (especially central venous dialysis and in-hospital dialysis) used to treat kidney failure is expensive. Additionally, there are concerns that the increasing life expectancy of the general population, improved treatments for the causes of kidney failure, such as diabetes, and the increased life expectancy of these patients will put pressure on total healthcare costs [62]. Furthermore, dialysis has a fundamental economic impact on society by taking away time and energy from patients and increasing the risk of unemployment. In the United States, over 75% of patients are unemployed at the start of dialysis [63]. It is hoped that the maturation of kidney organoids will provide an alternative to renal replacement therapy, thereby compensating for the economic losses caused by kidney disease and its treatment. Moreover, the development of kidney organoids can be extended to the research and development of non-renal diseases and therapies. For example, kidney organoids have the potential to accelerate pharmaceutical research since drugs are ultimately excreted by the kidneys. Drug excretion in the kidney involves three processes: glomerular filtration, tubular excretion, and tubular reabsorption, and different drugs have varying effects on these processes [64]. Furthermore, clearance is affected by the presence or absence of renal disease and the maturity of nephrons, leading to substantial variations in drug excretion even among patients receiving the same drug. Mature kidney organoids are expected to be of great benefit to the pharmaceutical industry by enabling drug excretion and toxicity studies to be conducted under in vitro conditions using human organ models [65,66,67].

## 5. Conclusions

Kidney organoids hold great promise as a tool for understanding the pathophysiology of kidney failure and as a potential therapeutic option for patients worldwide. However, current techniques for growing these organoids often result in an inadequate vascular network, limiting their clinical applications. To address this issue, three main strategies have emerged for enhancing the growth and vascularization of kidney organoids. First, transplantation into immunocompromised animals allows for host blood flow to promote organoid vascularization and maturation. Second, organ-on-chip technologies provide a platform for external regulation of mechanical and chemical factors during organoid culture. Third, decellularized extracellular matrix (ECM) from animals provides a micro niche for kidney organoid maturation and appears to contain unknown factors that promote vascularization. However, as efforts to improve vascularization have progressed, new challenges have emerged, including defects in the collecting ducts, chimerism, insufficient numbers of nephrons, and tumorigenesis. Further research is necessary to overcome these limitations and unlock the full potential of kidney organoids.

## Figures and Tables

**Figure 1 biology-12-00503-f001:**
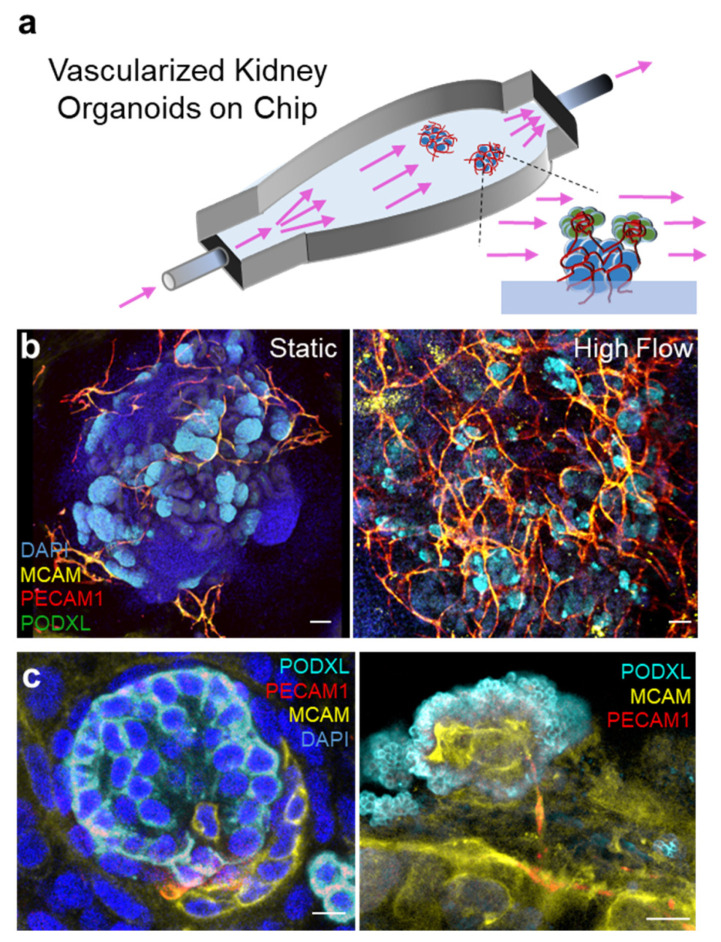
Vascularized kidney organoids cultured on millifluidic chips. (**a**) An illustration showing the experimental set up of the organoids-on-chip system. Arrows: fluid flow. (**b**) Whole organoid images showing glomeruli (PODXL) and vessels (MCAM/PECAM1) in static and perfused organoids. (**c**) High magnification images demonstrating vessel invasion into organoid glomeruli in perfused kidney organoids. Images adapted with permission from [14].

**Table 1 biology-12-00503-t001:** Comparison of transplantation experiments.

Cell Origin	Culture Prior to Graft (Day)	Animal	Site	Duration (Day)	Ref.
hiPSCs	7 + 18	Mouse	Kidney capsule	28	[12]
ESCs+ hiPSCs	5 + 16	Chicken	CAM	5	[23]
ESCs	7 + 11–12	Chicken	Coelomic cavity	1–8	[24]
hiPSCs	N/S	Mouse	Kidney capsule	14	[16]
hiPSCs	N/S	Mouse	Jejunal lymph nodes	21	[21]
hiPSCs	3 + 18	Mouse	Kidney capsule	7–42	[25]
hiPSCs	7 + 18	Mouse	Kidney capsule	14–21	[26]

## Data Availability

Not applicable.

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
