# Peer review of "Strategies for Improving Vascularization in Kidney Organoids: A Review of Current Trends"

_biology, 2023, doi:10.3390/biology12040503_

Round 1

Reviewer 1 Report

Konoe et al. well summarized the details of vascularization of kidney organoids and 3Ddiscusses the current challenges and efforts in developing matured and vascularized organoids. I have just minor concerns.

1.     3D bioprinting may be an attractive strategy for the vascularization and maturation of kidney organoids. Adding the advantages and pitfall of this strategy in the manuscript may give the insight to the readers.

2.     It would be better to add how the vascularized kidney organoids will be applied in the research field as well as in the industrial field.

3.      It would be better to add the pictures of the vascularized kidney organoids with the various magnifications.

Author Response

Reviewer #1:

Konoe et al. well summarized the details of vascularization of kidney organoids and 3Ddiscusses the current challenges and efforts in developing matured and vascularized organoids. I have just minor concerns.

  1. 3D bioprinting may be an attractive strategy for the vascularization and maturation of kidney organoids. Adding the advantages and pitfall of this strategy in the manuscript may give the insight to the readers.

Thank you for the insightful comment. We have added these advantages and pitfalls in Discussion.

  1. It would be better to add how the vascularized kidney organoids will be applied in the research field as well as in the industrial field.

One new paragraph is now added to the end of Discussion. This new paragraph discusses how vascularized kidney organoids would be applied in the research and industrial fields.

  1. It would be better to add the pictures of the vascularized kidney organoids with the various magnifications.

Vascularized organoid images are now added as a new figure. Thank you so much for the valuable comments.

Reviewer 2 Report

Two extremely well known authors review strategies to overcome the most persistent problem in application of organoid technology to medicine: by the time the organoid has begun to self-organize into the components of an organ subunit, such as the nephron, the organoid has often become large enough that diffusive transport of oxygen and nutrients is limiting and insufficient.

The authors review in vivo implantation, OoC, and the role of decellularized matrix.   

The manuscript is quite good and quite timely.  I have a few comments

- the in vivo section is fine, but refers to immune problems a little too sparingly

- The section on OoC doesnt really discuss vascularization; it delves more into the roles of FSS on differentiation.  Theres not a lot pulled from the soft lithography field and advances in capillary engineering from, for example, oxygenator design.

- the section on matrices is also a little thin and to an extent avoids the fundamental question that plagues the field- the millimeter to micron transition.   yes, dECM does encourage vascular growth, but.... at what stage will you embed an organoid? How will you control differentiation of the organoid once its is placed? What is the consequence and what are the limits of ischemia to the organoid while one is awaiting vaasculogenesis?

There are a wealth of papers outside of the organoid field that might be useful to discuss.  Long before organoids, the islet transplant community wrestled with this exact issue.   Attempted solutions are varied and in some cases completely unexpected- such as "shooting" carbon nanotubes through islets and lavaging explanted lungs with islets or dissociated islets.   The plastic surgery field as well may have contributions.  Brown and Humes published a paper in ASAIO J about simply encapsulating a rodent artery with a cassette into which serous fluid accumulated followed by stromal cells and vessels.  

At a high level I think the manuscript struggles to focus- is it about organoid tissue integration, organoid delivery, or narrowly about organoid vascularization?  Any is fine, but choose and stick to it.

Author Response

Reviewer #2:

Two extremely well known authors review strategies to overcome the most persistent problem in application of organoid technology to medicine: by the time the organoid has begun to self-organize into the components of an organ subunit, such as the nephron, the organoid has often become large enough that diffusive transport of oxygen and nutrients is limiting and insufficient.

The authors review in vivo implantation, OoC, and the role of decellularized matrix.  

The manuscript is quite good and quite timely.  I have a few comments

- the in vivo section is fine, but refers to immune problems a little too sparingly

- The section on OoC doesnt really discuss vascularization; it delves more into the roles of FSS on differentiation.  Theres not a lot pulled from the soft lithography field and advances in capillary engineering from, for example, oxygenator design.

- the section on matrices is also a little thin and to an extent avoids the fundamental question that plagues the field- the millimeter to micron transition.   yes, dECM does encourage vascular growth, but.... at what stage will you embed an organoid? How will you control differentiation of the organoid once its is placed? What is the consequence and what are the limits of ischemia to the organoid while one is awaiting

There are a wealth of papers outside of the organoid field that might be useful to discuss.  Long before organoids, the islet transplant community wrestled with this exact issue.   Attempted solutions are varied and in some cases completely unexpected- such as "shooting" carbon nanotubes through islets and lavaging explanted lungs with islets or dissociated islets.   The plastic surgery field as well may have contributions.  Brown and Humes published a paper in ASAIO J about simply encapsulating a rodent artery with a cassette into which serous fluid accumulated followed by stromal cells and vessels. 

At a high level I think the manuscript struggles to focus- is it about organoid tissue integration, organoid delivery, or narrowly about organoid vascularization?  Any is fine, but choose and stick to it.

Thank you for taking the time to review our manuscript and for providing insightful comments that have helped us to improve the focus on vascularization in our work. We have carefully considered reviewer’s suggestions and made significant revisions to our manuscript. We have now added three new paragraphs to the discussion section, which provide more detailed information on varied approaches for vascularization in kidney organoids. We have also elaborated a paragraph describing the 3D printer which could be used to enhance vascularization and maturation of organoids. Furthermore, we have included a new paragraph that discusses the extracellular matrix (ECM) in relation to vascularization. This section also discusses the potential challenges of the use of ECM for vascularization. Finally, we have added a new paragraph that explores the potential applications of vascularized kidney organoids. This section discusses the exciting possibilities for using our methodology to develop new treatments for kidney diseases, as well as for drug discovery and testing. Overall, we believe that our revised manuscript now clearly and effectively addresses the key topic of vascularization in kidney organoids. Thank you once again for your helpful feedback, and we hope that you find our revised manuscript to be both informative and compelling.

Round 2

Reviewer 2 Report

The authors have very gracefully accomodated my suggestions and I think the MS is stronger for it.  I hope they thought my comments helpful.